# Propagation of goose primordial germ cells in vitro relies on FGF and BMP signalling pathways
Dadakhalandar Doddamani[1], Bence Lázár[2,3], Kennosuke Ichikawa [1], Tuanjun Hu[1,4], Lorna Taylor[1], Elen Gócza[3], Eszter Várkonyi [2] & Mike J. McGrew [1] ✉

Mitotically active embryonic reproductive cells, the primordial germ cells (PGCs), are an ideal cell type for cryopreserving functional reproductive cells for avian species. Their low number in the avian embryo, however, renders cryopreservation and germline transmission methodologies difficult. Here, we develop a culture medium for the long-term in vitro culture of PGCs from the goose, *Anser anser domesticus*. In contrast to chicken, goose PGC self-renewal is dependent on the TGF-β family member, BMP4, and, conversely, is inhibited by Activin A. An RNA transcriptome analysis reveals commonalities between cultured PGCs from chicken and goose species, including a marked transcriptional difference between male and female goose PGCs. In vitro propagated goose PGCs are amenable to genetic modification using DNA transposons and colonising the gonads of xenogeneic sterile host embryos. These data demonstrate that the conservation and cryopreservation of the genetic diversity of the >1400 endangered bird species using PGCs remains a valid possibility.

Alongside efforts to sample the genetic diversity of extant bird species, the cryopreservation of somatic and germ cells from endangered avian species is imperative to aid the ex situ conservation programmes aimed at safeguarding bird genetic diversity[1–4]. Reproductive male cells in the form of differentiated spermatozoa can be collected and successfully cryopreserved for many vertebrate species[5]. Oocytes, however, are fewer in number and particularly difficult to cryopreserve for animals containing large amounts of yolk in the oocyte such as fish and bird species. Embryonic reproductive cells, the primordial germ cells (PGCs), are an ideal cellular resource for preserving the genetic diversity of species as they are mitotically active and differentiate into spermatogonia and oocytes after transplantation into host gonads of chimeric animals[6]. In birds, the PGCs are present in the pregastrula embryo and can be easily isolated from the forming circulatory system. Avian PGCs can be easily re-introduced into the circulatory system of the embryo and will subsequently form functional male and female gametes in the adult surrogate hosts[7]. Their low numbers (several hundred) in the avian embryo, however, limits their usefulness as a cellular resource. In contrast, the biobanking of somatic cells has been pursued for the last 30 years as it is anticipated that frozen somatic cells could be used for the cloning of endangered and extinct animals[4]. Encouraging progress has been made in reprogramming somatic cells into iPS cells and from there into

PGC-like cells. It is hoped that PGC-like cells will be differentiated into functional gametes[8]. However, in some bird species, a germ cell-restricted chromosome is present which is hypothesised to play a functional role during gametogenesis[9–11]. Conserved somatic cells from these bird species could never form a functional gamete. Thus, for many bird species, only the direct preservation of germ cells will be sufficient for the conservation and future formation of functional gametes[12].

PGCs isolated from mammalian and fish embryos have proven difficult to propagate in vitro[13–15]. Similarly, PGC-like cells derived from pluripotent mammalian cells can only be maintained for a limited time in culture[16]. The long-term culture of vertebrate PGCs was first achieved in an avian species, *Gallus gallus*[17]. Chicken PGCs were cultured in a medium containing growth factors, animal sera and a feeder cell layer and maintained expression of germ cell markers and importantly contributed to the germline of surrogate hosts[17]. Subsequently, it was shown that chicken PGCs could be cultured in a serum-free medium containing the growth factors fibroblast growth factor 2 (FGF2), Insulin and Activin A and maintained germline competence. This previous study also demonstrated that the chicken PGCs can be grown in a medium containing either Activin A or bone morphogenetic protein 4 (BMP4), or both growth factors together[18]. The use of a transgenic chicken host containing an iCaspase9 transgene

[1]The Roslin Institute and Royal (Dick) School of Veterinary Studies, University of Edinburgh, Edinburgh, UK. [2]National Centre for Biodiversity and Gene Conservation, Institute for Farm Animal Gene Conservation, Gödöllő, Hungary. [3]Animal Biotechnology Department, Institute of Genetics and Biotechnology, Hungarian University of Agriculture and Life Sciences, Gödöllő, Hungary. [4]National Gene Pool of Waterfowl, Jiangsu Agri-Animal Husbandry Vocational College, Taizhou, China. ✉e-mail: mike.mcgrew@roslin.ed.ac.uk

expressed specifically in the developing germ cells permitted 100% transmission of chicken donor PGCs[19]. The development of chicken PGCs biorepositories and sterile surrogate hosts is aiding efforts to biobank breeds of chicken and the generation of genome-edited chickens for research[20–22].

The long-term propagation and manipulation of PGCs from bird species other than the chicken has not been achieved[23,24]. For Galloanserae, quail and duck PGCs can be cultured in vitro only for periods up to 50 days[25–27]. Similarly, for a Neoaves songbird, zebra finch PGCs can be cultured for short periods and modified to produce transgenic songbirds[28,29]. However, the limited in vitro proliferation of PGCs from non-chicken species has limited efforts to both biobank and genome edit other bird species. Here, we hypothesized that the modulation of transforming growth factor beta (TGF-β) growth factors will be sufficient for in vitro self-renewal of PGCs for other bird species and tested this using the domestic goose.

## Results

### Propagation of goose PGC cultures from embryonic blood

Chicken PGCs can be propagated in semi-suspension in a serum-free medium containing B27 supplement, insulin, chicken ovotransferrin, and FGF2 containing additional Activin A (FAOT) or BMP4 (FBOT) alone, or both growth factors together (FABOT). The lowering of calcium levels from the physiological level of 1.8–0.15 mM reduced cell-cell adhesion and permitted the continuous in vitro culture of the more adherent female chicken PGCs. Starting with FAOT or FBOT medium, we first screened for PGC growth using pooled embryonic blood from quail, guinea fowl, duck, turkey, or goose eggs. For these species, we achieved an initial 24 h of outgrowth of putative PGCs. Subsequently, these cells either clumped together or adhered to the surface of the well and differentiated into fibroblast-like cells. However, from goose embryonic blood and FBOT medium, we observed the continuous outgrowth of suspension cells resembling PGCs (Fig. 1A). Cultured 'PGC'-like cells formed adherent floating clumps with some morphological characteristics similar to chicken PGCs: the presence of cytoplasmic lipid vacuoles and small cell surface protrusions. We next produced a variation of this medium containing additional FGF1, insulin-like growth factor 1 (IGF1), plus the vitamin retinol, and the sterol lipid cholesterol. Our hypothesis was that using two growth factors to activate the FGF and IGF pathways, would have better success in activating the insulin/IGF and FGF signalling pathways in divergent vertebrate species. In addition, the supplementation of cholesterol would aid in maintaining the characteristic intracellular lipid vesicles of PGCs.

Embryonic blood from stage 16 Hamburger Hamilton (HH) equivalent goose embryos (72 h incubation) was cultured in 1× B27 supplement, FGF1 and FGF2, IGF1, retinol, cholesterol, and 0.15 mM calcium and supplemented with either BMP4 or Activin A. The cells were fed once every three days over a four-week period. Embryonic blood supplemented with Activin A did not generate PGC outgrowth. In contrast, cells supplemented with BMP4 obtained numbers greater than 50,000 cells within 4 weeks (Fig. 1A–D). These putative PGCs were cultured for three months before cryopreservation (Fig. 1E, F). Immunofluorescent staining demonstrated that these cells expressed the germ cell markers DAZL, but were not positive for SSEA1 (Supplementary Fig. 1). We named this medium composition containing 1× B27 supplement, FGF1 and FGF2, BMP4, IGF1, retinol, cholesterol, and 0.15 mM calcium, Goose medium.

To measure the proliferation rate of the putative goose PGCs, we cultured male and female goose PGCs (approximately 200 PGCs) in the Goose medium for a period of 20 days (Fig. 1G). The doubling time for the female and male PGCs was 2.1 days and 1.7 days, respectively (Fig. 1H) and the number of male PGCs was significantly higher than the female PGCs (p-value: 0.01). The in vitro doubling time of goose PGCs is greater than that for chicken PGCs (doubling time ~ 21 h)[18,21].

### Activin A inhibits and FGF promotes goose PGC proliferation

We next investigated the effect of FGF and TGF-β signalling molecules and FGF on goose PGC self-renewal. Our initial efforts to establish the goose PGC culture failed in the presence of Activin A. To delineate the effect of Activin A on goose PGC culture, we performed a proliferation experiment on goose PGCs in Goose medium with or without additional Activin A. The cells grow well in the BMP4 containing Goose medium (Fig. 2A) but the addition of Activin A to Goose medium inhibited PGC proliferation (Fig. 2B and Supplementary Fig 2). Likewise, the removal of FGF1 and FGF2 from the Goose medium showed that the addition of FGF is necessary for in vitro self-renewal (Fig. 2C).

### Surface attachment is needed for goose PGC self-renewal

Both chicken and goose PGCs do not adhere to the well substrate when cultured but instead rest loosely on the bottom of culture wells, detaching when gently pipetted[18,30]. To determine if PGC interaction with a surface is necessary for outgrowth, we cultured the cells on either standard tissue cell culture plates or non-coated polypropylene plates. Tissue cell culture polystyrene plates are treated to neutralise positive charges, increasing non-specific interaction between the plate surface and the cell[31]. Surprisingly, we observed that goose PGCs in the Goose medium did not proliferate in a non-adherent cell culture plate (p-value: 0.029) indicating that interaction with a surface matrix is needed for goose PGC self-renewal (Fig. 3A).

### Cholesterol is required for goose PGC propagation

As eukaryote cell membranes contain lipids, sterols and sphingolipids, the addition of cholesterol supports the growth of many mammalian cell lines[32]. To determine the necessity of cholesterol in the culture media, we performed a proliferation experiment in a Goose medium with and without cholesterol. We cultured 200 goose germ cells in the Goose medium and counted cell numbers after 10 days in culture. This analysis demonstrated that the goose PGCs required cholesterol for increased propagation in vitro (Fig. 3B).

### Impact of CaCl₂ levels on goose PGC cultures

Our previous study has shown that the chicken PGCs in the FAOT medium tend to adhere, form clumps and require a low concentration of calcium in a medium along with the addition of heparin[18]. We established Goose PGC culture with 0.15 mM $CaCl_2$, we presumed goose PGCs require a low concentration of calcium. To examine, the aspired blood sample from the goose embryos of stage 15–17 HH was split into two wells containing Goose medium, one with media with 0.15 mM and the other with 0.075 mM of $CaCl_2$. After 4-weeks of culturing, goose PGCs in lower $CaCl_2$ grew to a higher number than those growing in a normal low $CaCl_2$ medium (Fig. 3C). Using these conditions, we attempted to derive PGC cultures using blood cultured from single stage 16 HH equivalent Embden goose embryos. We were able to derive male and female PGC lines with a derivation rate of 75% (Table 1).

### Comparison of human IGF1 and Turkey IGF1 on goose PGCs

Goose medium contains a high concentration of insulin (in B-27 supplement) and also additional human Insulin-like Growth Factor 1 (IGF1) at a concentration of 50 ng/ml. Both IGF1 and insulin bind to receptor IGF1R and activate several signalling pathways related to cell proliferation, differentiation, regeneration, and growth. However, human IGF1 proteins share 83% homology with the goose IGF1 protein and turkey IGF1 share 98% homology with the goose IGF1 protein, suggesting IGF1 may have species-specific signalling activity. We performed a comparative study of human and turkey IGF1 on goose PGC growth rate in the absence of insulin. The outcome of the analysis reveals that there is no significant difference in the proliferation rate for IGF1 proteins (Supplementary Fig. 4), suggesting additional IGF1 is not needed for goose PGC self-renewal.

### Culture of PGCs from an independent domesticated breed of geese

We subsequently attempted to replicate our findings in an independent research facility using independently sourced reagents and eggs from a different breed of geese. We obtained fertile eggs from Hungarian Frizzled geese and repeated the derivation experiment from embryonic blood. The Goose medium differed by lacking the addition of retinol, due to the

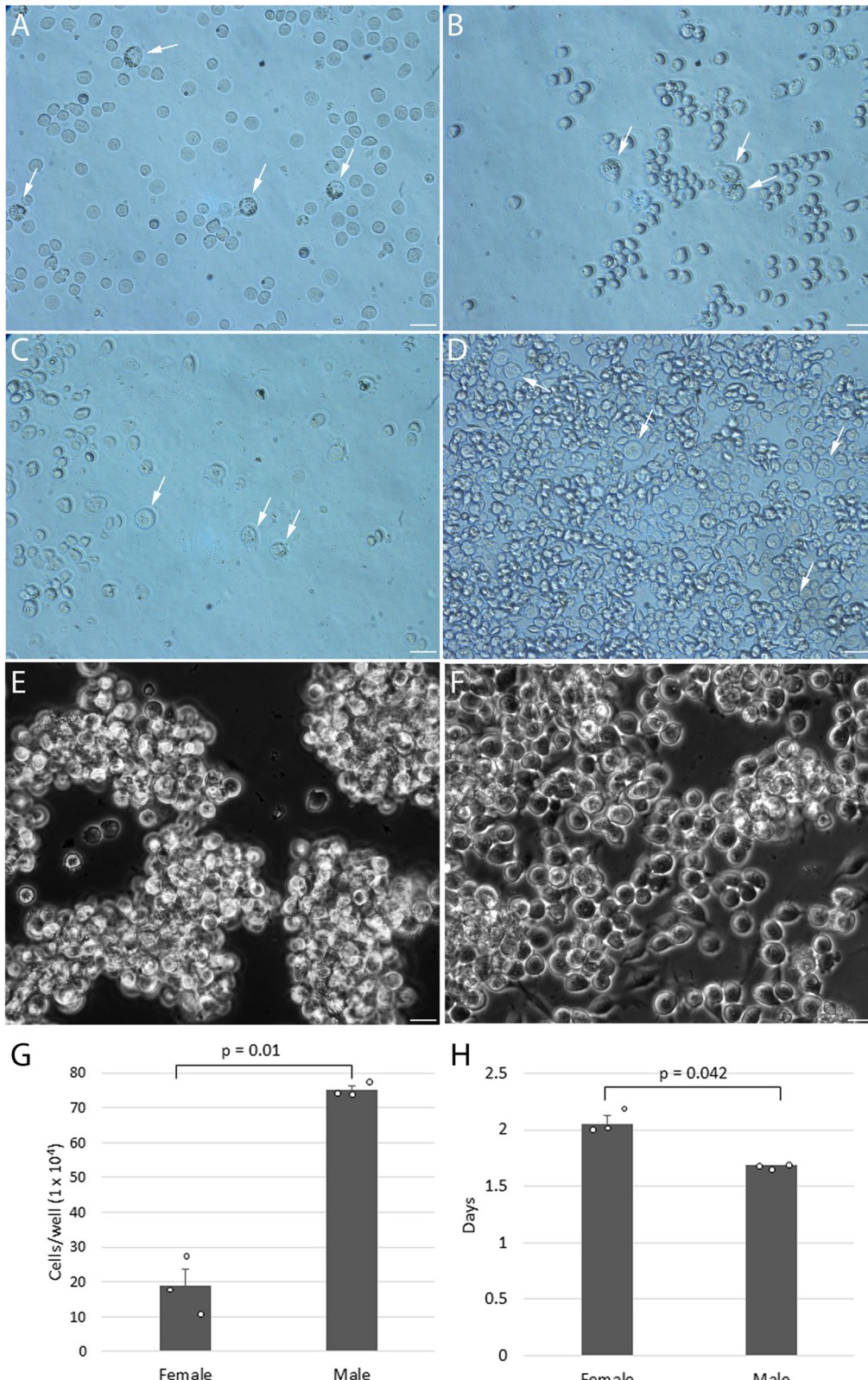

**Fig. 1 | In vitro culture of goose PGCs in a medium containing BMP4. A–D**. Time course of goose PGCs cultured from embryonic blood. **A** 24 h. **B** 48 h. **C** 72 h. **D** 96 h. Goose PGCs have a round morphology and are larger in size than blood cells. **E** Male goose PGCs cultured for more than three months. **F** Female goose PGCs cultured for more than three months. Size bars = 20 μm. **G** A growth rate measurement for male and female goose PGCs was carried out by culturing approximately 200 PGCs in a goose medium for 20 days. Cell counts showed a significantly slower female PGC growth compared to male PGCs. **H** Doubling time was calculated based on the initial and final population of PGCs over 20 days of culturing. The doubling time for the male PGCs is significantly lower than that of female PGCs ($p$-value: 0.042). $n = 3$ independent replicates using one male and one female cell line. Error bars = SEM.

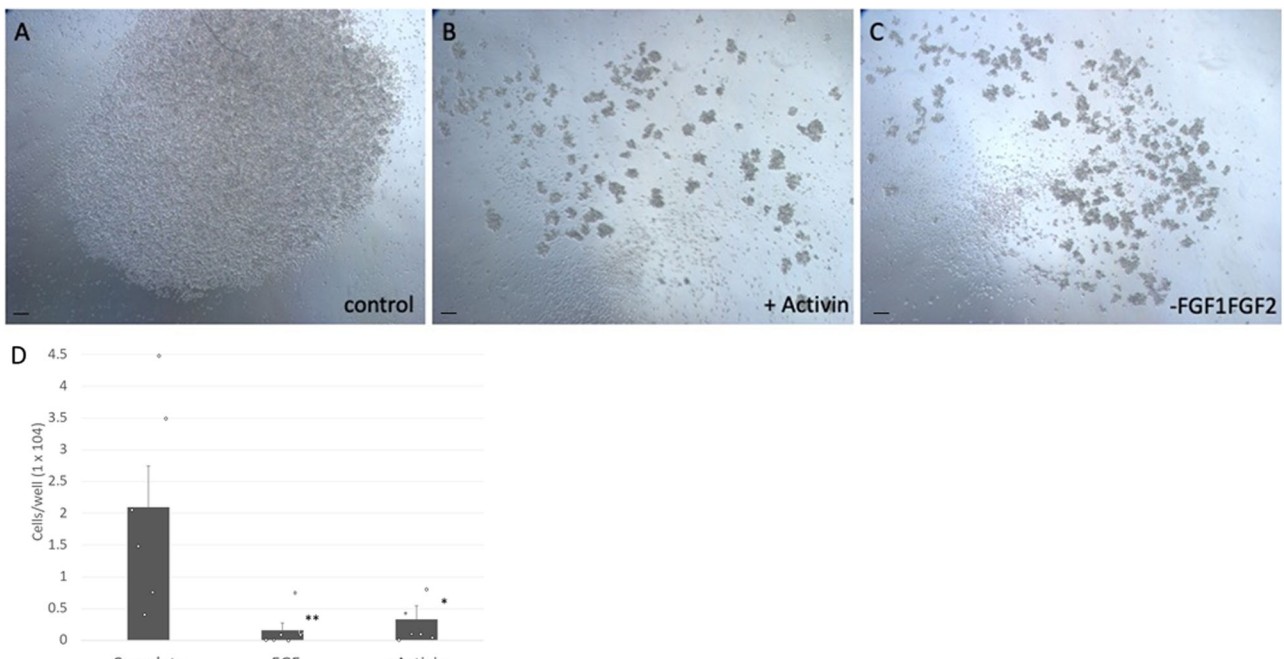

**Fig. 2 | FGF is required for PGC propagation, and is inhibited by Activin. A** The PGCs were grown in an established media condition (FBOT). Two hundred PGCs were plated in a well and then counted after 14 days in culture. **B** The addition of Activin A inhibits cell proliferation. **C** Absence of growth factors FGF1 and FGF2 inhibited cell proliferation. **D** FGF was removed from the culture medium or Activin A (25 ug/ml f.c.) was added. $n = 6$ independent experiments. Error bars, SEM. $^*p = 0.017$, $^{**}p = 0.009$. Scale bars, 150 μm.

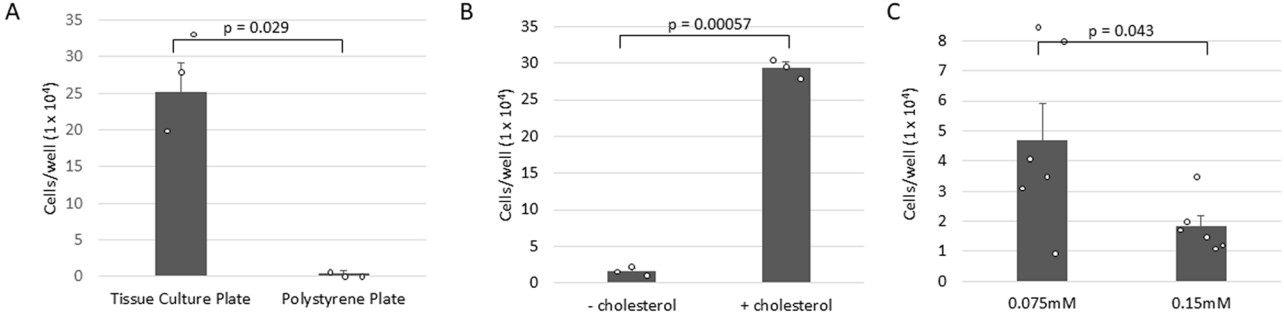

**Fig. 3 | Characteristics of cell culture conditions on in vitro proliferation. A** Goose PGCs were grown in normal and non-adherence culture plates. The non-adherence tissue culture plates do not support cell growth. **B** Goose PGCs were grown in medium with and without cholesterol. **C** Goose PGCs were grown in two concentrations of calcium to reduce cell adherence. The lower concentrations of calcium significantly improve cell proliferation. All experiments were repeated three to six times.

unavailability of this reagent, and the use of osmo-reduced commercial DMEM, and 0.075 mM $CaCl_2$ (see "Materials and Methods", Supplementary Table 1). We successfully established cultures from 15 of the 36 eggs sampled (Table 1). Cell numbers were between 76,000 and 640,000 PGCs per derivation (embryo) at the time of cryopreservation (Table 2). To summarise these results, we found that goose PGCs proliferated in a medium containing BMP4, but with a slower doubling time than was previously observed for chicken PGCs. The addition of cholesterol and lowering of calcium levels aided goose PGC propagation. These data show that goose PGCs from single embryos can be propagated by in vitro culture to provide sufficient cell numbers for cryopreservation.

### Expression of germ cell-specific genes is conserved in chicken and goose PGCs
In a previous study using RNA transcriptome, we identified a set of genes that showed exclusive expression in chicken PGCs in comparison with the other cell line transcriptome data[33]. The known PGC markers like *DDX4* and *DAZL* confirmed their expression specific to germ cells (Supplementary

Fig. 4A). The transcriptome data of chicken PGCs and goose PGCs are comparable (Supplementary Fig. 5). Using the goose RNA transcriptome data, we examined the expression of this set of germ cell-specific genes. As expected, the germ cell-specific genes expressed in goose PGCs, and their expression level are comparable with that of chicken PGCs (Supplementary Fig. 6A). We confirmed the expression of the known *DAZL* and *DDX4* in cultured PGCs using primers. All chicken and goose-derived samples were positive for *DDX4, DAZL* and *GAPDH* expression (Fig. 4A, C, and D).

### Analysis of pluripotency gene expression in cultured chicken and goose PGCs
A previous comparison of transcriptome data of chicken PGCs, pluripotent and non-pluripotent cells revealed that chicken PGCs express many pluripotency genes. For example, chicken PGCs express the pluripotency genes *PRDM14, NANOG*, and *POU5F1*[33,34]. The expression pattern of *PRDM14* is not conserved in mammalian PGCs as mouse PGCs express *PRDM14*, unlike human PGCs which do not express *PRDM14*. We analysed the expression of other pluripotency-related genes in the goose PGCs. Due to a

**Table 1 | Derivation rates for goose PGC cultures**

| Breed | No. of eggs (*) | No. of blood isolations | No. of successful PGC cultures$ (%) | No. of male PGC cultures | No. of male cultures/male eggs sampled (%) | No. of female PGC cultures | No. of female cultures/female eggs sampled (%) | Derivation rate (%) |
|---|---|---|---|---|---|---|---|---|
| UK Embden goose | 22 (2) | 12 | 9 (75%) | 6 | 6/8 (75%) | 3 | 3/4 (75%) | 9/12 (75%) |
| Hungarian frizzled goose | 90 (3) | 36 | 15 (42%) | 9 | ND# (60.0%) | 6 | ND# (40.0%) | 15/36 (42%) |

*Number of independent experiments.

$Embden goose cultures were counted between 4 and 5 weeks, successful cultures had >30,000 cells. Frizzled goose cultures were counted at the time of cryopreservation (4–8 weeks post-blood sampling).

#ND not determined. The goose embryo was not retained for sex PCR. Successful PGC cultures were sexed before cryopreservation.

**Table 2 | Cryopreserved Hungarian frizzle PGCs**

| Freezing date | Cell line ID | Sex | Viability | Cells/ cryotube | Number of cryotubes |
|---|---|---|---|---|---|
| 24/05/2021 | 3 | Male | 91.13% | 412,200 | 4 |
| 14/05/2021 | 4 | Female | 90.74% | 260,100 | 4 |
| 31/05/2021 | 5 | Female | 93.64% | 145,800 | 4 |
| 02/06/2021 | 7 | Female | 82.50% | 358,200 | 4 |
| 07/05/2021 | 9 | Male | 70.40% | 226,500 | 4 |
| 07/05/2021 | 10 | Male | 72.68% | 516,750 | 4 |
| 26/04/2021 | 11 | Male | 84.48% | 188,438 | 4 |
| 14/05/2021 | 12 | Female | 90.39% | 305,400 | 4 |
| 26/04/2021 | 16 | Male | 67.56% | 121,563 | 4 |
| 12/05/2021 | 17 | Female | 88.61% | 640,800 | 4 |
| 26/04/2021 | 24 | Male | 54.88% | 76,250 | 4 |
| 24/05/2021 | 26 | Male | 81.15% | 273,600 | 4 |
| 28/05/2021 | 29 | Male | 90.25% | 542,250 | 4 |
| 19/05/2021 | 31 | Female | 72.32% | 228,600 | 4 |
| 31/05/2021 | 33 | Male | 89.03% | 100,800 | 4 |

PGC cultures were frozen after 1–2 months in culture.

lack of complete genome annotation for the goose, a subset of chicken genes does not have orthologs in this species, which hinders the comprehensive cross-species analysis. However, this analysis reveals that goose PGCs express the core pluripotency factors *NANOG, LIN28, SOX21, PRDM14, DNMT3B, KLF5, GRHL1,* and *VRTN* (Fig. 4B and Supplementary Figs. 4B and 6B). To summarise, goose PGCs express many germ cell-specific and pluripotency genes comparable to chicken PGCs.

We further confirmed the expression of *PRDM14* in cultured PGCs. Our analysis finds positive expression of *PRDM14* in both male and female goose PGCs and confirms the conserved expression of *PRDM14* in avian PGCs (Fig. 4B and Supplementary Figs. 4B and 6B).

### TGF-β signalling pathways are active in cultured chicken and goose PGCs

Previous research showed that the activation of the TGF-β signalling pathway genes in the chicken PGCs is requisite for their in vitro propagation[18]. In chicken PGCs, both Activin and BMP signalling pathways were active. To explore the type of TGF-β signalling pathways that are active in goose PGCs, we looked for the expression of TGF-β type I, II, and III receptors in the transcriptome data. The transcriptome analysis depicts the members of type I, II and III receptors expressed in avian PGCs (Supplementary Fig. 7). In addition, we also found that the FGF receptors *FGFR1, FGFR2* and *FGFR3*, insulin receptors *IGF1R* and *IGF2R*, and retinol receptor *RARRES1* are expressed in chicken and goose PGCs. However, due to incomplete reference genome assembly for the goose, a subset of chicken genes does not have identified orthologs in the goose species.

### Differential gene expression between male and female goose PGCs

To understand the changes in the molecular network between male and female PGCs, we compared the transcriptome of goose male and female PGCs. The current dataset contains three male and two female PGC libraries of the goose. The transcriptome comparison reveals that the male and female PGCs are transcriptionally distinct (Fig. 5A). The differential expression study reveals that 235 genes are expressed higher in male PGCs with a log2fold change value higher than one, an average of normalised expression in males higher than 500 and an adjusted *p*-value lower than or equal to 0.05 (Fig. 5B, C and Supplementary Data 2).

In contrast, 116 genes are expressed higher in the female PGCs with a log2fold change value lower than −1, an average of normalised expression in

**Fig. 4 | Verification of pluripotency gene expression.** RNA from chicken PGC lines 12 F and 20B and goose PGC lines G3, and G7 were used as a template for RT-PCR analysis of gene expression. **A** *DDX4*, **B** goose *PRDM14* specific primers, **C** *DAZL*, and **D** *GAPDH*. A day one of incubation goose embryo was used as a PCR control. − lane, one day goose embryo, −RT; + lane, one day goose embryo, +RT.

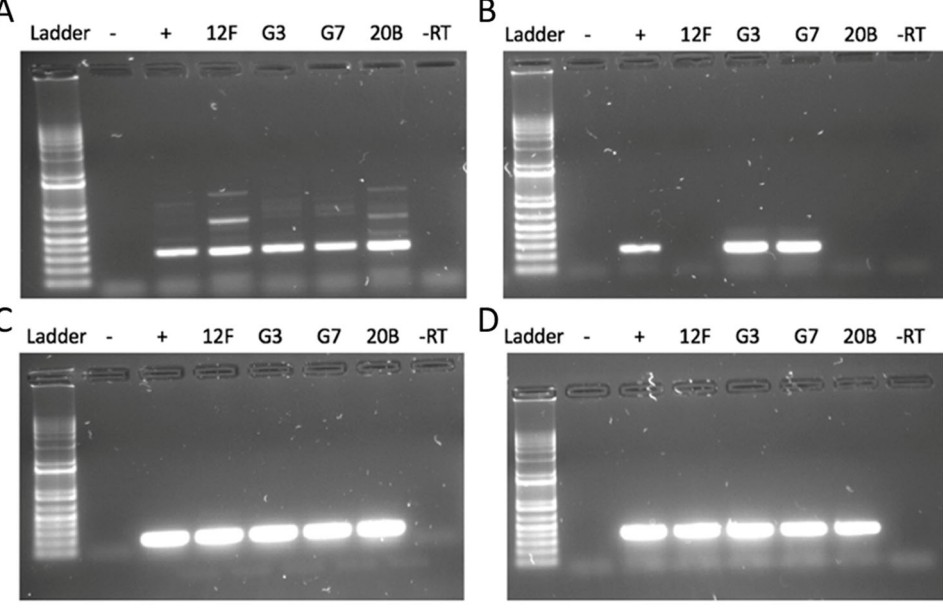

females higher than 500 and an adjusted *p*-value lower or equal to 0.05 (Supplementary Data 3). Genes related to cell adhesion (FDR value 2.00E-03), cell−cell junction assembly (FDR 1.23E-02), cell growth (5.35E-03), adherent junction organization (FDR 3.02E-03) (Fig. 5D) such as *CD44*, *CDH11*, *CDH4*, *COL12A1*, *CTGF*, *FLRT2*, *ITGA11*, *ITGA8*, *ITGB8*, *LAMA2*, *NRP1*, *POSTN*, *SPON2*, *TGFBI*, *TGFB2*, and *THBS2*, signal transduction (FDR value 2.66E-02) related genes *TGFB1*, *TGFB2*, are expressed relatively higher in female PGCs (Fig. 5B, D and Supplementary Data 4). The gene *GAS2* which is involved in the ovulation process and *LGR5* which is involved in the oocyte differentiation process are expressed higher in female goose PGCs. The female PGCs are enriched in the integrin signalling pathway (FDR value 3.03E-05) and Inflammation mediated by chemokine and cytokine signalling pathway (FDR value 2.26E-02) (Supplementary Data 5).

### Stable genetic modification of goose PGCs using DNA transposons

To follow goose PGC migration *in ovo*, we labelled the putative PGCs with a fluorescent reporter gene. We and others previously showed that chicken PGCs can be efficiently modified using DNA transposons[35,36]. We transfected goose PGCs with a piggyBac vector containing a CAG puromycin reporter construct containing a GFP or RFP transgene. PGCs were cultured and selected with puromycin over one month until a pure GFP+ or RFP+ population was obtained (Fig. 6A–C).

### Migration of goose PGCs to xenogeneic chicken gonads

To demonstrate the germ cell functionality of the goose PGCs, we assayed the migration of the cultured goose PGCs from the circulatory system to the genital ridge. As geese are both seasonal breeders and scarce, we used both chicken and duck eggs as xenogeneic hosts for PGC injections. Cultured GFP+ goose PGCs were injected in the dorsal aorta of day 2.5 chicken eggs and incubated until embryonic day 6-14. We observed that GFP+ cells colonised the gonad of chicken host embryos (Supplementary Fig. 9). To further test the xeonogenic migration of avian PGCs, GFP+ chicken PGCs were injected in the dorsal aorta of day 3 duck embryos and incubated until embryonic day 6. We observed that GFP+ cells colonised the gonad of host duck embryos (Supplementary Fig. 8). We next tested the migration and colonisation of goose PGCs in host chicken embryos ablated for endogenous germ cells. Heterozygote iCaspase9 embryos were co-treated with B/B compound during injection with goose PGCs. We found that the ablation procedure did not interfere with host colonisation by goose donor male and female PGCs (Fig. 6). These data show that cultured avian PGCs will efficiently colonise the embryonic gonads from other bird species.

### Discussion

Previous studies described cell culture media that facilitated the long-term in vitro propagation of chicken PGCs[17,18]. This development led to efforts to propagate PGCs from other bird species[24,27,28]. However, these media conditions only supported PGC growth for short periods in vitro (two to four weeks), suggesting that the long-term propagation of avian PGCs from bird species other than chicken would not be possible. Here, we demonstrate a serum-free culture medium to support long-term in vitro self-renewal of goose PGCs (Fig. 7).

As the initial effort to culture goose embryonic blood in Activin A-containing medium failed, we hypothesised that goose PGCs may grow in a medium containing a second TGF-β signalling molecule. Consequently, our attempts to culture goose PGCs containing BMP4 were successful with a derivation rate of 42–75% which is comparable to the published derivation rate for chicken PGCs of ~68%[1,18]. Our results showed that the rate of proliferation of goose PGCs is slower compared to that of chicken PGCs which are grown in a FAOT medium and showed that once derived, goose PGCs proliferated faster in a medium without additional antioxidants or retinol. This overall proliferation difference may be due to the physiological differences between goose and chicken embryos; the goose embryo grows slower than a chicken embryo, hatching after 27 days of incubation in comparison to 21 days for the chicken.

It is worth noting that this is the second bird species that did not require SCF for in vitro self-renewal of migratory stage embryonic PGCs, suggesting the migratory stage of PGC development does not depend on SCF ligand.

Previous studies showed that the TGF-β signalling pathway genes are active in chicken PGCs a constitutively active SMAD3 protein could replace Activin A or BMP4 in a culture medium whereas constitutively active SMAD5 protein could not[18]. As Activin A inhibited Goose PGC propagation, there appears to be a fundamental difference between TGF-β signalling and PGC proliferation in these two bird species. It should be noted that our experiments carried out on cultured PGCs are not equivalent to the de novo derivation of PGC cultures from blood. The lack of available eggs due to the seasonal breeding of geese did not allow us to test further derivations under multiple medium conditions.

Future research will test if xenogeneic gamete formation in sterile surrogates is possible. Researchers have previously shown the germ line transmission of chicken PGCs that underwent successful spermatogenesis in the testes of guinea fowl and duck[37,38]. Also, pheasant PGCs successfully completed gametogenesis and formed functional spermatozoa in chicken[39]. Thus, it is likely that male goose PGCs will be able to form functional gametes in a male chicken host. However, female oogenesis depends on

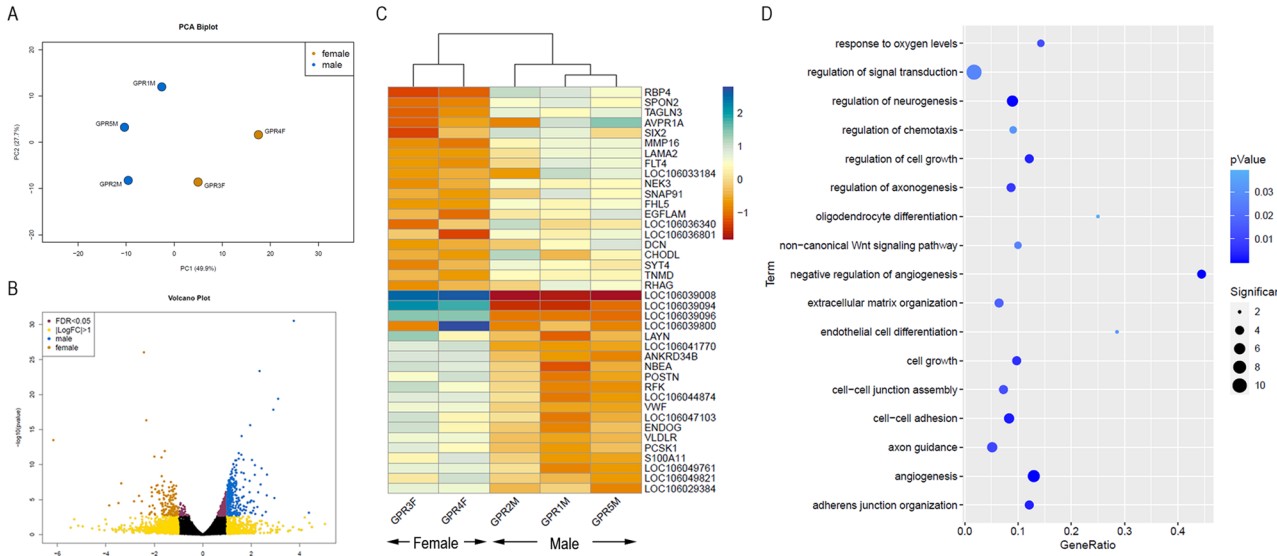

**Fig. 5 | Transcriptome comparison between male and female PGCs. A** The PCA plot based on the 1000 highly variable genes shows that male and female PGCs are transcriptionally distinct. **B** The volcano plot shows the differentially expressed genes between male and female PGCs. **C** Heatmap shows the expression of the top 20 DEGs between male and female PGCs. **D** The graph shows the gene ontology enrichment of female DEGs. The cell adherence, cell-cell junction and cell growth-related genes enriched.

multiple interactions between the hypothalamus-pituitary and the oocyte and the surrounding somatic follicular granulosa cells and may not be possible in xenogeneic hosts.

## Materials and methods
### Derivation and culturing of goose PGCs
Goose eggs were obtained from small commercial breeders. Embden commercial white goose eggs were provided by the farm of Mary Pocock (www.Fertileeggs.uk, Gloucester, UK). Hungarian Frizzled goose eggs were obtained from the poultry facility of the National Centre for Biodiversity and Gene Conservation, Institute for Farm Animal Gene Conservation (Göd-öllő, Hungary).

Goose-fertile eggs were incubated for 70–72 h to attain a morphological stage equivalent to HH16. One to two microliter of blood was isolated from the dorsal aorta using a 1.0 mm pulled glass capillary and added to 300 µl of PGC culture medium. Goose medium, containing 4 ng/ml FGF1, 4 ng/ml FGF2, 25 ng/ml BMP4, and Ovotransferrin (50 µg/ml) (FBOT), B27 (+) insulin (Thermofischer Scientific), Vitamin (1x), B12, Cholesterol, Retinol, and IGF-1(50 µg/ml), 0.15 mM calcium chloride DMEM (Thermofisher Scientific). Firstly, the goose PGCs were cultured in a 48-well plate, and the medium was changed every 3 days. Approximately four weeks later, PGCs were transferred to a 24-well plate, culturing medium was changed every 2 days for approximately four months. Stem-Cellbanker (Amsbio) cryopreservation medium was used as the freezing medium to mix with the cultured goose PGCs. Each blood derivation was frozen in four vials, labelled and stored at −80 °C in the freezer overnight. The vials were transferred to −150 °C on the next day for permanent storage.

### RNA isolation
Five goose PGC lines consisting of three male and two female cell lines were cultured for an additional three months during which cell pellets were isolated every seven days. The cell pellets were collected in RNA Later and RNA was isolated from pooled pellets using an RNeasy Plus Micro Kit (Qiagen). The quality of the RNA was estimated using Nanodrop and an Agilent RNA tape station 2200 instrument. Samples with RIN values greater than seven were considered for sequencing (Supplementary Fig. 9). The library preparation and sequencing were carried out at the Edinburgh Genomics sequencing facility (Edinburgh). Illumina paired-end reads of length 150 base pairs were generated.

## RNA analysis
### Generation of RNAseq data and quality filtering.
The paired-end RNA sequencing data of read length 150 basepair were generated from five goose PGC lines including three male and two female cell lines. The NGS-Qcbox[40] was used for quality filtering of raw reads and adaptor sequences were trimmed using Trimmomatic software[41]. The trimmed reads were mapped using HiSAT2[42] with the default parameters on the draft goose genome (goose: GooseV1.0)[43]. The feature count software[44] was used to count the number of reads aligned to genes. The read count matrix was used in the DESeq2 package[45] to identify differential gene expression. The following criteria were applied to identify differentially expressed genes (DEGs):

Male DEGs: |Log2FC| > 1, $p$-value < 0.05 and average of normalised expression in male cells > 500

Female DEGs: |Log2FC| > 1, $p$-value < 0.05 and average of normalised expression in female cells > 500

The gene ontology enrichment analysis was performed in the Panther database[46]. The differentially expressed genes were used as input by considering the chicken gene list as a background and searching against the GO slim and Panther pathways databases. Fisher's exact is used for the $p$-value and the calculate false discovery rate option is used for FDR value estimation.

An average of 150 million paired-end RNASeq data from the cultured goose PGCs was generated. The adaptor-trimmed reads were mapped to the reference genome (v1). Overall, an average of 87% of reads were mapped to the goose reference genome. The reference-based transcriptome assembly from one of the female samples generated 345 K contigs with greater than 500 bp length and 67 Kb maximum length contig. However, a reference annotation file was used to count the number of reads mapped to the genomic features.

### Reverse transcription PCR
A 15 µl total volume reaction mixture for PCR was made of 2 µl 1:10 diluted sample cDNA, 0.3 µl 10 mM dNTPs (Invitrogen), 0.3 µl 50 pmol/µl primers (forward and reverse), 1.5 µl 10× buffer + MgCl$_2$, 0.1 µl Fast Start Taq and 10.5 µl H$_2$O. The reaction was carried out at 94 °C for 5 min, followed by 35 cycles of 94 °C for 30 s, 50 °C for 30 s, 72 °C for 1 min, and a final extension of 72 °C for 5 min. Samples were run on a 1% TAE agarose gel. PCR products were sequenced to verify DNA products.

**Fig. 6 | Goose PGC genetic modification and migration in chicken host embryos.**
**A** GFP expressing male Goose PGCs after modification with piggyBac GFP transposon. Chicken embryo injected with male GFP⁺ goose PGCs at ED2.5 and assayed at ED5. Bar, 100 um. **B** iCaspase9 surrogate host (ED14) treated with B/B compound and injected with male RFP⁺ Goose PGCs. Bar, 200 um. **C** iCaspase9 surrogate hosts (ED14) were treated with B/B compound and injected with female RFP⁺ Goose PGCs. Bar, 200 um.

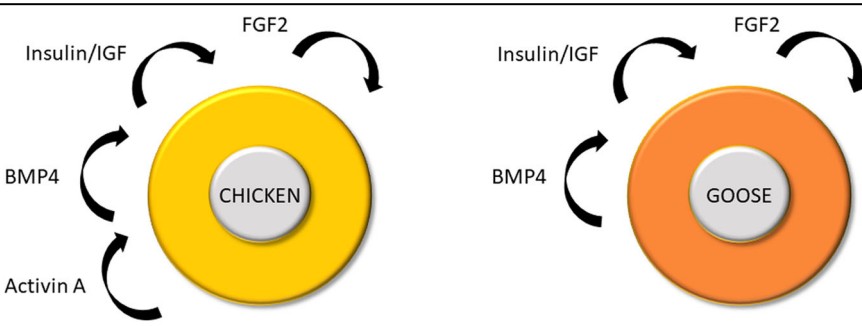

**Fig. 7 | Model of chicken PGC and goose PGC self-renewal.** Chicken PGCs self-renewal requires FGF2, insulin/IGF, and Activin A or BPM4. In contrast, goose PGC self-renewal requires FGF2, insulin/IGF, and BPM4.

## PGC transfections

Goose PGCs were transfected using chicken PGC conditions[47]. Briefly, 1 µl of CAG piggyBac transposase plasmid and 1 µg of piggyBac CAG-GFP-IRES-puro or piggyBac CAG-GFP-IRES-mCherry transposon plasmid were transfected into 100,000 goose PGCs using Lipofectamine 2000 in Opti-MEM I medium (Life Technologies). Four days post-transfection, cells were selected in a Goose medium containing 0.1 µg/ml puromycin for approximately 25 days.

## Immunofluorescence staining

Fluorescence imaging was performed on goose PGC to test the expression of specific PGC antigens. The cells were fixed in 4% PFA for 20 min and then permeabilized with 0.1% Triton X-100 for 15 min. After blocking in 2% goat serum for 30 min, the cells were incubated with rabbit anti-DAZL (1:250 dilution, Abcam) and mouse anti-SSEA-1 (1:250 dilution, R&D system) antibody cocktail for 1 h, washed for 1 h in PBT, followed by AF488-conjugated anti-rabbit (1:500 dilution, Life Technologies) and AF594-conjugated anti-mouse IgM (1:1000 dilution, Life Technologies) antibodies for 30 min. Cells were mounted with DAPI and imaged for fluorescence.

## Sex determination of embryos and RT PCR primers

The sex of the sampled embryo was determined by using the 'W' chromosome-specific primers[48,49].

## Gonad migration assay

A goose PGC solution was prepared at 5000 cells/µl using the avian KO DMEM containing 0.075 uM CaCl2 and Fast Green (Sigma) and 0.5 mM B/B Homodimerizer (TaKaRa, Japan). Approximately 2 µl of the PGCs solution was injected into the dorsal aorta of embryos of the E2.5 (stage 16-16+ HH) iCaspase9 chicken line[19] using a glass capillary. After the injection, 50 uL of PBS containing Penicillin/Streptomycin and 15 uM of the B/B Homodimerizer was pipetted on top of the embryo, and then, the eggs were incubated at 37.5 °C for a total of 14 days. Gonads of the embryos incubated were observed under a fluorescence microscope (AXIO Zoom.V16; Zeiss, Germany) and photographed with an AxioCam HRm camera (Zeiss).

## Statistics and reproducibility

Statistical analysis of biological replicates was performed by unpaired two-tailed $t$-test (Figs. 1 and 3) and Tukey HSD test (Fig. 2). Sample size and definition of replicates are given in the figure legends. Goose PGC derivations were conducted independently in Hungary and the UK using independently sourced reagents.

## Primer sets

GAPDH: 650 bp product
CAGATCAGTTTCTATCAGC, TGTGACTTCAATGGTGACA
GOOSE PRDM14 (407 bp)
TCTTCTGTACGGACCCCATC, GCACTGGATAGCCGTCAAAT
DAZL (455 bp)
CTGCACCGCAATTCCATAGTG, TTTTCTGAAGTGATGCGCCC
DDX4 (406 bp)
GCGTGGATGGCTAACTCTGG, TGCCACGCAGAGGACTATTT
Sex PCR
PRIMER SET 1 (CLINTON, 2001) (200 bp, female only)
GAAATGAATTATTTTCTGGCGAC,
CCCAAATATAACACGCTTCACT
PRIMER SET 2 ([48]) (150 bp, 160 bp) GAGAAACTGTGCAAAACAG,
TCCAGAATATCTTCTGCTCC

## Reporting summary

Further information on research design is available in the Nature Portfolio Reporting Summary linked to this article.

## Data availability

Goose circulatory PGC RNA-Seq data sets generated in this study are publicly available in the NCBI SRA database under accession number "PRJNA1000303". All data are available from the corresponding author upon reasonable request. Numerical source data for all graphs and analyses in the manuscript can be found in Supplementary Data File 1.

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

## Acknowledgements

The authors thank the Pocock farm for providing fertile goose eggs and duck eggs, Luer Chen for generating preliminary data[50], and the NARF staff for maintaining the iCaspase9 chicken line. This work was supported by Institute Strategic Grant Funding from the BBSRC (BBS/E/RL/230001C and BBS/E/RL/230002A) to the Roslin Institute. KI was funded by the Japan Society for the Promotion of Science (JSPS) Overseas Research Fellowship. EV was funded by VEKOP-2.3.2-16-2016-00012.

## Author contributions

Conceptualization: D.D., B.L., M.J.M., and T.H. Methodology: L.T., K.I., B.L., and M.J.M. Investigation: M.J.M., E.G., E.V., L.T., K.I., B.L., T.H., and D.D. Supervision: M.J.M., E.G., and E.V. Writing: M.J.M., E.G., E.V., K.I., B.L., T.H., and D.D.

## Competing interests

M.J.M. is an inventor on patent application WO 2020074915 for the iCaspase9 transgenic chicken. The University of Edinburgh is the applicant. The remaining authors declare no competing interests.

## Institute review board statement

We have complied with all relevant ethical regulations for animal use. All UK experiments and procedures were performed in accordance with relevant UK Home Office regulations. Experimental protocols and studies were reviewed by the Roslin Institute Animal Welfare and Ethical Review Board (AWERB) Committee. None of the experiments in this work were at the protected stages of development under the Animal Welfare Act (1986). For the experiments in Hungary, animals were kept and maintained according to general animal welfare prescriptions of the Hungarian Animal Protection Law (1998; XXVIII). All experimental methods were approved by the Institutional Ethics Review Board of the Institute for Farm Animal Gene Conservation (no. 7/2011).
