## [Transparent Peer Review file · Communications Biology]

Propagation of goose primordial germ cells in vitro relies on FGF and BMP signalling pathways

Corresponding Author: Professor Mike (J) McGrew

Version 0:

Reviewer comments:

Reviewer #1

(Remarks to the Author)

In this paper, Doddamani et al. of McGrew's group successfully cultured the goose PGCs in vitro long-term period. The authors elucidate essential factors for goose PGCs culture and introduced foreign genes into these cultured goose PGCs. Furthermore, the authors demonstrate migration and colonization of cultured goose PGCs into chicken genital ridge, which showed that the cultured goose PGCs were genuine avian germ cells. In my knowledge, this is the first case of stable long-term culture of PGCs in avian species except chicken. Also, these results could give the scientific clues for how we culture the avian germ cells, maintaining germ cell characteristics for long periods, and which factors are essential to maintain germness and self-renewal of germ cells. I think this paper could contribute to avian germ cell biology, avian species conservations and avian genome editing.

Specific comments

1. The mixed use of the terms 'Activin A' and 'Activin' can cause confusion for readers. Providing a clear distinction between the two would be helpful for readers.
2. Please unify figure citation. For example, in line 104, please reword (Figure 1a) to (Figure 1A).
3. Full name of specific words should be given such as FGF, BMP4 and HH stage.
4. Figure 1G: It is better to describe the period of culture after seeding of 200 PGCs in figure legends.
5. What is the meaning of x days in Figure 2 legend?
6. Figure 4: Please describe which sample was used as negative (-) control.
7. Line 134 and 215: Is it proper to use word "avian PGC" ? If the results of this paper are commonly applied to all avian species, it would be proper. I think it should be carefully re-considered using this word.
8. Line 300-304: Please add proper references for this description.

Reviewer #2

(Remarks to the Author)

Doddamani et al. modified the culture medium composition established by Whyte et al. to optimize conditions for goose PGC culture. They found that removing Activin from the FAot medium, which is suitable for chicken PGCs, and adding BMP4 instead supported the proliferation of goose PGCs. Additionally, they observed that the addition of cholesterol, an adherent culture substrate, and changes in calcium levels further enhanced PGC proliferation. RNAseq analysis showed that the gene expression profile of cultured goose PGCs was comparable to that of cultured chicken PGCs. They also examined the effects of sex and strain. Moreover, they demonstrated that cultured goose PGCs could be genetically modified and that transplantation experiments confirmed that these cultured PGCs retained the ability to home to the gonads. This paper marks the first successful establishment of a long-term culture method for PGCs in avian species beyond chickens, representing a groundbreaking achievement. The insights gained from this study could accelerate advancements in avian PGC culture techniques. While further study is required to determine whether cultured goose PGCs can form gametes and produce subsequent generations, this remains a significant question of interest. Although the paper is well-composed, several revisions could improve it. Below, I outline these suggested modifications:

Major comments:

Regarding Supplementary Figure 1, is there a possibility of autofluorescence in the SSEA-1 staining? SSEA-1 signals are typically membrane-localized; however, in this figure, the signal appears cytoplasmic and resembles DAZL's pattern. SSEA-1 may lack cross-reactivity with goose PGCs. To my knowledge, SSEA-1 does not recognize duck PGCs and shows low cross-reactivity with emu and quail PGCs as well.

Line 130: It is unclear at this point whether cholesterol, FGF1, and retinol are included in FBOT. FBOT appears multiple

times in the Results sections, but the medium's exact composition is hard to follow.

Line 161: The final culture conditions reportedly yield a 75% PGC establishment efficiency, yet there's no mention of how cholesterol or retinol supplementation affects this efficiency. While cholesterol appears to accelerate the growth rate of cultured goose PGCs, readers may benefit from knowing whether it also increases PGC derivation from blood. Including any insights on the effects of retinol supplementation would also be valuable.

Line 178: Providing a clear reason for omitting retinol from the PGC cultures derived from Hungarian Frizzled geese would be helpful to readers.

Minor comments:

Line 47: "Embryonic germ cell" may bring associations with EG cells; it might be more appropriate to change this term. Using "embryonic reproductive cell," as mentioned in the abstract, could work well.

Line 52: Replacing "fertile laid eggs" with "embryos" might be more accurate.

Line 68: A reference should be added to support the statement, "Similarly, PGC-like cells derived from pluripotent mammalian cells can only be maintained for a limited time in culture."

Line 102: "Figure 1a" should be "Figure 1A."

Line 144-149: Since this section discusses cholesterol's effects, it doesn't align well under the title "Surface attachment is needed for avian PGC self-renewal." It would be better to separate this into a distinct section.

Stage description: The HH stage notation varies throughout, appearing as "stage OOHH," "stage OO HH," "HHOO," and "HH stage OO." This should be standardized. The same applies to "TGF- β " and "TGF β ."

Line 302: The authors mention previous findings showing successful spermatogenesis of chicken PGCs in the testes of guinea fowl and duck, and successful gametogenesis of pheasant PGCs, resulting in functional spermatozoa in chicken. Appropriate references should be provided for these claims.

Line 365: "0.3 μ l 50 pmol/ μ l primer (I, ii)" – Clarify what "I, ii" represents.

Line 370: "Goose PGCs were transfected as chicken PGCs (ref)." – Please include the missing reference.

Line 382: Does "HH stage 16-16" contain a typo?

Line 390: Only one GAPDH primer seems to be shown, possibly due to a missing comma.

Reviewer #3

(Remarks to the Author)

In this study, the cultivation method for goose PGC is described. The optimal condition for various PGC depends on species and sometimes even on strains, therefore, the successful cultivation and the differences from the method for chicken PGC are interesting and valuable.

1. The fate of xeno-transplanted goose PGC toward sperm or oocyte is to be updated, if possible.
2. How about the availability of this new method for PGC of quail, guinea fowl, duck, and turkey?
3. Please include the information on cytokines and chemicals in Supplementary Table 1.

Version 1:

Reviewer comments:

Reviewer #1

(Remarks to the Author)

The authors addressed the reviewer's comments well.

Reviewer #2

(Remarks to the Author)

I think that the revised manuscript has sufficiently addressed all my comments.

There is an extra "and" in Line 75, so please delete it.

Reviewer #3

(Remarks to the Author)

My questions are answered appropriately, and I have no further questions.

We thank the reviewers for their time to review our manuscript. We appreciate that they view that this research may 'accelerate avian PGC culture techniques' and 'contribute to avian germ cell biology, avian species conservations and avian genome editing'. We have re-submitted the manuscript with changes tracked. We found additional errors not noted by the reviewers in the Figure legends and in the text and have also highlighted these changes.

Reviewer 1

Specific comments

1. The mixed use of the terms 'Activin A' and 'Activin' can cause confusion for readers. Providing a clear distinction between the two would be helpful for readers.

We assayed the ligand Activin A for goose PGC culture. We have now used the term Activin A throughout the manuscript, except for the discuss of Activin and BMP signalling pathway in the discussion (line 218). This is now the only section that does not specify Activin A.

2. Please unify figure citation. For example, in line 104, please reword (Figure 1a) to (Figure 1A).

We have corrected this mistake and have checked the manuscript for other errors.

3. Full name of specific words should be given such as FGF, BMP4 and HH stage.

We have added the full names to the abbreviations when they first appear in the text of the manuscript: FGF, BMP, IGF, HH, and TGF- β .

4. Figure 1G: It is better to describe the period of culture after seeding of 200 PGCs in figure legends.

These cells were grown for 20 days. This information was added to the figure legend.

5. What is the meaning of x days in Figure 2 legend?

We forgot to add the length of the culture period. It was 14 days and this information is now added to the figure legend.

6. Figure 4: Please describe which sample was used as negative (-) control.

The figure legend is incorrect, - and + refer to reverse transcriptase, the sample used for both is incubation day 1 goose embryo RNA. This was added to the figure legend.

7. Line 134 and 215: Is it proper to use word "avian PGC" ? If the results of this paper are commonly applied to all avian species, it would be proper. I think it should be carefully re-considered using this word.

Thank you for this comment.

We currently believe that the need for matrix attachment will vary between bird species. We have changed this sentence to 'Surface attachment is needed for goose PGC self-renewal'.

The same reasoning applies to TGF- β pathway ligands. This may not be true for all avian species. We have changed this sentence to' TGF- β signalling pathways are active in cultured chicken and goose PGCs'.

8. Line 300-304: Please add proper references for this description.

We have added the three references lacking from this section. (Kang, van de Lavoie, Liu)

Reviewer 2:

Major comments:

1. Regarding Supplementary Figure 1, is there a possibility of autofluorescence in the SSEA-1 staining? SSEA-1 signals are typically membrane-localized; however, in this figure, the signal appears cytoplasmic and resembles DAZL's pattern. SSEA-1 may lack cross-reactivity with goose PGCs. To my knowledge, SSEA-1 does not recognize duck PGCs and shows low cross-reactivity with emu and quail PGCs as well.

The reviewer is correct. We have repeated the experiment using no primary antibody controls. The SSEA1 immuno-staining is no greater than the background from the secondary. The commercial DAZL antibody works well and stains the goose PGCs. We have replaced the supplementary figure with the new data and controls and changed the text in the results section.

2. Line 130: It is unclear at this point whether cholesterol, FGF1, and retinol are included in FBOT. FBOT appears multiple times in the Results sections, but the medium's exact composition is hard to follow.

We should not use the term FBOT. FBOT was only used only for the initial experiment on line 97. Afterwards we added FGF1, retinol, IGF1, and cholesterol to the medium. This composition is called goose medium. We have now given this definition for Goose medium on line 122, and used the term Goose medium for the remainder of the paper.

3. Line 161: The final culture conditions reportedly yield a 75% PGC establishment efficiency, yet there's no mention of how cholesterol or retinol supplementation affects this efficiency. While cholesterol appears to accelerate the growth rate of cultured goose PGCs, readers may benefit from knowing whether it also increases PGC derivation from blood. Including any insights on the effects of retinol supplementation would also be valuable.

One of our major limitations is the inability to obtain fertile goose eggs. We therefore cannot repeat the derivations using Goose medium +/- cholesterol +/- IGF1, and +/- retinol. The Hungarian derivations did not use Retinol which suggest addition of retinol is not necessary.

4. Line 178: Providing a clear reason for omitting retinol from the PGC cultures derived from Hungarian Frizzled geese would be helpful to readers.

We know that the Hungarian group could not source retinol for their experiments and did not include this supplement in their Goose medium. We have added this (trivial) reason for omitting retinol to the results section (line 183) We have listed the two formulations for Goose medium in Table 1.

Minor comments:

1. Line 47: "Embryonic germ cell" may bring associations with EG cells; it might be more appropriate to change this term. Using "embryonic reproductive cell," as mentioned in the abstract, could work well.

We welcome this suggestion and have changed the text.

2. Line 52: Replacing "fertile laid eggs" with "embryos" might be more accurate.

We have re-written this sentence. 'Avian PGCs can be easily re-introduced into the circulatory system of the embryo and will subsequently form functional male and female gametes in the adult surrogate hosts (Park et al., 2019)'.

3. Line 68: A reference should be added to support the statement, "Similarly, PGC-like cells derived from pluripotent mammalian cells can only be maintained for a limited time in culture."

We have referenced Ohta, 2017 here.

4. Line 102: "Figure 1a" should be "Figure 1A."

This error was also noted above by Reviewer 1.

5. Line 144-149: Since this section discusses cholesterol's effects, it doesn't align well under the title "Surface attachment is needed for avian PGC self-renewal." It would be better to separate this into a distinct section.

*We have added a new section entitled '**Cholesterol is required for goose PGC propagation**'*

6. Stage description: The HH stage notation varies throughout, appearing as "stage OOHH," "stage OO HH," "HHOO," and "HH stage OO." This should be standardized. The same applies to "TGF- β " and "TGF β ."

We correct this and gave the definition to the abbreviation HH on line 113. We have now used TGF- β throughout the manuscript.

7. Line 302: The authors mention previous findings showing successful spermatogenesis of chicken PGCs in the testes of guinea fowl and duck, and successful gametogenesis of pheasant PGCs, resulting in functional spermatozoa in chicken. Appropriate references should be provided for these claims.

Reviewer 1 also asked for this correction (above).

8. Line 365: "0.3 μ l 50 pmol/ μ l primer (I, ii)" – Clarify what "I, ii" represents.

This sentence was corrected to 0.3 μ l 50 pmol/ μ l primers (forward and reverse)

9. Line 370: "Goose PGCs were transfected as chicken PGCs (ref)." – Please include the missing reference.

This sentence and reference were corrected. 'Goose PGCs were transfected using chicken PGCs conditions (Idoko-Akoh and McGrew, 2023).'

10. Line 382: Does "HH stage 16-16" contain a typo?

Yes, this was corrected to 'stage 16-16' HH'

11. Line 390: Only one GAPDH primer seems to be shown, possibly due to a missing comma.

We have corrected the primer sequences.

Reviewer 3

1. The fate of xeno-transplanted goose PGC toward sperm or oocyte is to be updated, if possible.

We are pursuing grant funding and animal regulatory authorisation to be able to transplant the goose PGCs into the sterile iCaspase9 surrogate host and hatch chicks. We currently do not have authorisation to carry out this experiment. We decided to publish this research now as it would be useful to other lab-based avian conservation groups.

2. How about the availability of this new method for PGC of quail, guinea fowl, duck, and turkey?

This medium is almost sufficient to culture duck PGCs. However, we only achieved the derivation of one duck PGC culture, from more than 50 duck eggs, so we did not include this information.

3. Please include the information on cytokines and chemicals in Supplementary Table 1.

We have added this information to Table 1.